# Identification of Beneficial Microbial Consortia and Bioactive Compounds with Potential as Plant Biostimulants for a Sustainable Agriculture

**DOI:** 10.3390/microorganisms9020426

**Published:** 2021-02-19

**Authors:** Silvia Tabacchioni, Stefania Passato, Patrizia Ambrosino, Liren Huang, Marina Caldara, Cristina Cantale, Jonas Hett, Antonella Del Fiore, Alessia Fiore, Andreas Schlüter, Alexander Sczyrba, Elena Maestri, Nelson Marmiroli, Daniel Neuhoff, Joseph Nesme, Søren Johannes Sørensen, Giuseppe Aprea, Chiara Nobili, Ombretta Presenti, Giusto Giovannetti, Caterina Giovannetti, Anne Pihlanto, Andrea Brunori, Annamaria Bevivino

**Affiliations:** 1Department for Sustainability, ENEA, Italian National Agency for New Technologies, Energy and Sustainable Economic Development, Casaccia Research Center, 00123 Rome, Italy; silvia.tabacchioni@enea.it (S.T.); cantalec@gmail.com (C.C.); antonella.delfiore@enea.it (A.D.F.); alessia.fiore@enea.it (A.F.); giuseppe.aprea@enea.it (G.A.); chiara.nobili@enea.it (C.N.); ombretta.presenti@enea.it (O.P.); andrea.brunori@enea.it (A.B.); 2AGRIGES srl, 82035 San Salvatore Telesino, Italy; stefania.passato@agriges.com (S.P.); patrizia.ambrosino@agriges.com (P.A.); 3Center for Biotechnology (CeBiTec), Bielefeld University, 33615 Bielefeld, Germany; huanglr@cebitec.uni-bielefeld.de (L.H.); aschluet@cebitec.uni-bielefeld.de (A.S.); asczyrba@cebitec.uni-bielefeld.de (A.S.); 4SITEIA.PARMA, Interdepartmental Centre for Food Safety, Technologies and Innovation for Agri-Food and Department of Chemistry, Life Sciences and Environmental Sustainability, University of Parma, 43124 Parma, Italy; marina.caldara@unipr.it (M.C.); elena.maestri@unipr.it (E.M.); nelson.marmiroli@unipr.it (N.M.); 5Department of Agroecology & Organic Farming, Rheinische Friedrich-Wilhelms-Universität Bonn, 53121 Bonn, Germany; jhett@uni-bonn.de (J.H.); d.neuhoff@uni-bonn.de (D.N.); 6Department of Biology, University of Copenhagen, Universitetsparken 15 Bldg 1, 2200 Copenhagen, Denmark; joseph.nesme@bio.ku.dk (J.N.); sjs@bio.ku.dk (S.J.S.); 7Centro Colture Sperimentali, CCS-Aosta S.r.l., 11020 Quart, Italy; giustogiovannetti@hotmail.com (G.G.); caterina@micosat.it (C.G.); 8Natural Resources Institute Finland (Luke), Myllytie 1, 31600 Jokioinen, Helsinki, Finland; anne.pihlanto@luke.fi

**Keywords:** SIMBA, sustainable agriculture, plant growth-promoting microorganisms, microbial consortia, metagenome fragment recruitments, delivery methods, in vitro compatibility, bioactive compounds

## Abstract

A growing body of evidence demonstrates the potential of various microbes to enhance plant productivity in cropping systems although their successful field application may be impaired by several biotic and abiotic constraints. In the present work, we aimed at developing multifunctional synthetic microbial consortia to be used in combination with suitable bioactive compounds for improving crop yield and quality. Plant growth-promoting microorganisms (PGPMs) with different functional attributes were identified by a bottom-up approach. A comprehensive literature survey on PGPMs associated with maize, wheat, potato and tomato, and on commercial formulations, was conducted by examining peer-reviewed scientific publications and results from relevant European projects. Metagenome fragment recruitments on genomes of potential PGPMs represented in databases were also performed to help identify plant growth-promoting (PGP) strains. Following evidence of their ability to coexist, isolated PGPMs were synthetically assembled into three different microbial consortia. Additionally, the effects of bioactive compounds on the growth of individually PGPMs were tested in starvation conditions. The different combination products based on microbial and non-microbial biostimulants (BS) appear worth considering for greenhouse and open field trials to select those potentially adoptable in sustainable agriculture.

## 1. Introduction

Soil microbial applications are a supportive strategy for sustainable management fostering the reduction of chemical pesticides and synthetic fertilizers in agriculture [1,2]. Soil indigenous and artificially applied plant growth-promoting microorganisms (PGPMs), the so-called plant probiotics [3,4,5], show an important role in promoting soil fertility and enhancing plant health due to their ability to improve crop productivity and nutritional quality [6], as well as plants’ resistance to pathogens and environmental stresses [7] and plants’ tolerance to abiotic stresses [8]. They include a wide variety of microorganisms, such as arbuscular mycorrhizal fungi (AM-fungi), phosphate solubilizing microorganisms, nitrogen-fixing bacteria, PGP rhizobacteria, actinomycetes, biocontrol strains, and endophytic bacteria [9], and vary from broader soil beneficial microorganisms through to specialized epiphytes and endophytes [10]. These microorganisms possess the ability to promote plant growth both by indirect or direct mechanisms, as well as a combination of both. Indirect mechanisms include, e.g., the control of plant pathogens either by stimulation of the plant’s defense mechanisms or by their antagonistic suppression through the production of antibiotics and siderophores. In contrast, direct mechanisms refer, for example, to the stimulation of plant’s hormone production, such as auxins or cytokinins. Additionally, microbial mobilization of sparingly available nutrient sources, such as recalcitrant soil phosphates or the associative N_2_-fixation, also belong to the latter category [11]. Nevertheless, under open field conditions, numerous biotic and abiotic constraints may hinder their plant growth-promoting efficacy and reproducibility, limiting their successful use in agriculture [12]. The response to PGPM soil inoculation may also vary considerably depending on the specific microbe, plant species, soil type, inoculant density, and environmental conditions. In general, shortly after the microorganisms are released into the soil, the microbial population declines progressively. The prolonged survival of applied microorganisms and the persistence of their effectiveness are objectives to be pursued by scientific research.

The first objective when considering inoculation with PGPMs is to find the most efficient microorganisms available [11]. Next, a study of the specific inoculant formulation, i.e., a carrier and a microbial agent [13], is generally undertaken to counteract the stresses to be endured upon transfer to the competitive and often harsh soil environment, including, e.g., a well-established indigenous soil microflora [14]. Finally, the chosen formulation (the laboratory or industrial process of unifying the carrier with the bacterial strain in liquid, organic, inorganic, polymeric, or encapsulated formulation) and method of application [15], determine the chances of success of the inoculant preventing its rapid decline in the soil. Most approaches for plant growth promotion imply the use of single-strain inoculants as biofertilizers, while only few consider microbial consortia products, i.e., the combination of two or more microbial species [16]. Whenever a single strain should result ineffective to exert PGP effects in field, particularly under stressful conditions, the use of multi-strain inoculants could represent a valid strategy to increase community efficiency and promote plant growth [17]. Today, synthetic community of different microorganisms able to interact synergistically are currently being devised [3,17,18]. In vitro studies indicate that mixtures of microorganisms determine a synergic interaction by providing nutrients, removing inhibitory products, and stimulating beneficial physiological traits, such as nitrogen fixation. Compared to single-species inoculation, multi-species inoculum frequently, increases plant growth and yield, and improves the availability of minerals and nutrients, providing the plants with more balanced nutrition [16].

Although many microorganisms show good performance in specific trials, their application in the field is often not translated into effective PGP action due to the heterogeneous and unpredictable environment that often obstacles the establishment of inoculated microbes [19]. A better understanding the reasons for the failures in the field may lead to the development of bioinoculants that are effective in natural conditions. One possible strategy is the use of tailored-microbial consortia that can favour the PGPMs success upon inoculation and the interactions between PGPMs and soil abiotic and biotic (indigenous soil organisms) components [14,19,20]. Microbial consortia, when inoculated into the soil, can develop specific interactions at various scales of time and space (physical contact, chemical signaling, and metabolic exchange) leading to emergent properties, that is their combination is more than the sum of the parts. In practical terms, the choice of microbial mixtures with high functional diversity may help to improve the chances of success of the inoculant, preventing its rapid decline in the soil, that depends on how functional, complementary, and synergic the candidate PGPMs are [21]. Moreover, state-of-the-art microbiome analyses by applying methods of metagenome research offer new opportunities to study the performance of PGPM strains in soil and to discover new microorganisms featuring PGP properties. For example, deep metagenome sequencing for an agricultural Chernozem soil from long-term field experiment carried out in Germany revealed the presence of so far non-cultured species encoding putative PGP traits [22]. Likewise, soil metagenome sequence datasets deposited in nucleotide sequence databases can be used to address the question which PGP species are best adapted to a given soil type or habitat. For instance, metagenome fragment recruitments were carried out to follow the fate of the inoculant plant protective strain *Bacillus amyloliquefaciens* FZB42 brought into the rhizosphere of lettuce (*Lactuca sativa*) [23].

New frontiers in plant biostimulants could profit from the beneficial associations of microorganisms and natural-based compounds [24,25]. The interest in bioactive compounds in modern agriculture results from the trend to search for natural substances that can reduce the application of synthetic agrochemicals in agriculture, thus limiting the presence of chemical residues in foodstuff, in line with the Farm to Fork Strategy of the European Green Deal [26], and making the agriculture more sustainable and resilient. Bioactive compounds, like plant protein hydrolysates and other plant extracts, when applied in small quantities, may play an important role in crop production by promoting vegetative growth, nutrient uptake and the tolerance of plants to abiotic stresses not only directly through the activity of signaling molecules but also indirectly by changing the microbial community in the phyllosphere [27,28,29]. Therefore, they are promising candidate to improve the efficiency of microbial consortia and favor the crop production in an environmental-friendly manner. The identification of the best combination PGPMs-bioactive compounds will permit the development of a second generation of plant biostimulants (biostimulant 2.0) with specific synergistic action able to make agriculture more sustainable and resilient [24].

Within the frame of the Horizon 2020 SIMBA project (Sustainable Innovation of Microbiome Applications in the Food Systems), we aimed to identify compatible microbial single-strain inoculants with proven PGP potential to be used for the set-up of synthetic microbial consortia (MC) inoculants for maize, wheat, tomato and potato crops. The identification of beneficial microorganisms was carried out through the survey of a large number of articles and project reports on PGPMs under different biotic and abiotic stress conditions. PGP strains were also retrieved from Project Partners microbial collections. The compatible. PGPMs were used to set-up MC assuring the highest level of functional diversity, i.e., including at least one PGPM capable to express one of the specific desired traits of the consortia (i.e., nitrogen fixation, phosphorus solubilization, etc.). Then, large-scale metagenome fragment recruitments were done to determine natural occurrence and prevalence of selected PGPM strains in soil and rhizosphere habitats represented by metagenome sequence data that are publicly available in databases. Finally, the effect of agro-industrial and plant-derived protein hydrolysates on microbial growth of the single strains utilized in the formulation of MC was evaluated in vitro to define the best MC-biostimulant combination.

## 2. Materials and Methods

### 2.1. Literature Survey: Search Strategy and Data Collection

The first step of the identification of the suitable microorganisms, capable to exert a plant growth promoting effect on maize, wheat, tomato, and potato, following studies conducted either in greenhouse and/or open field, consisted of a literature survey. A systematic search of the peer-reviewed literature was carried out between September 2018 and January 2019 in the “Web of Science by Thomas Reuter”, “AGRICOLA by U.S. National Agricultural Library International coverage”, and “Google scholar” search engines, with the following keywords “Plant Growth-Promoting Rhizobacteria (PGPR)” or “biofertilizer” or “rhizobacteria” and “field” and “crop name (tomato, potato, wheat and maize)”. The whole WEB space was also searched by GOOGLE using the same keywords. The requirements to include the articles were (i) the experiments had to be carried out either in field or in greenhouse using natural soils, (ii) the experimental design had to be described in detail and to include replications and untreated controls, (iii) all replications and controls had to be managed in parallel under *ceteris paribus* conditions, (iv) results had to be available, including the treatment mean of yields, standard deviation (SD), and statistical significance. To speed up the process, whenever possible, the above-mentioned information was verified by reading the abstracts; otherwise, articles were downloaded/recovered and analyzed completely. In some cases, the reference list of particularly relevant publications was also checked to identify further studies. Furthermore, published results of other EU related projects, including BIOFECTOR (2012–2017, No. 312117, http://www.biofector.info, accessed on 31 January 2019) and VALORAM (2009–2014, No. 227522, https://www.ucc.ie/en/valoram/, accessed on 31 January 2019), were taken also into account to identify additional potential candidates.

With the aim of determining a ranking of the scientific articles based on the validity and reliability of the experimental procedures and obtained results, all data were extracted and compiled in an Excel file, creating individual worksheets for each crop and organized in a single database. The scientific articles were evaluated and discriminated according to the procedure described in Data Sheet S1. Only studies reaching a fixed threshold were considered for PGPMs selection.

To identify the state-of-the-art on carriers and delivery systems available nowadays, bibliographic searches were performed in the first half of 2019 on “Elsevier’s Scopus”, “Web of Science Clarivate Analytics”, and “Google Scholar”. The search was performed by looking specifically for delivery systems, such as vermiculite and zeolite, in combination with keywords containing “microorganisms”, “plant growth promoting”, “PGPM”. Considering the low number of hits, the search was not limited to the crop plants of interest for the project (maize, wheat, tomato, and potato) to recover a wider range of literature. Applications to non-agronomic contexts, e.g., remediation of contaminated sites, were also included to address the advantages and disadvantages of specific carriers. It has to be reported that many publications in this field were published on journals from India and China, or on books, making more difficult the recovery of papers. Reviews and meta-analyses were preferred as sources of information.

### 2.2. Metagenome Fragment Recruitments on Genomes of Candidate PGPM

For the fragment recruitment approach, the genomes of the selected PGP strains (see Appendix A) were downloaded from GenBank to serve as templates regarding metagenome sequence mappings. Secondly, appropriate soil, root, and metagenome datasets were identified in the ENA (European Nucleotide Archive, Cambridgeshire, UK, https://www.ebi.ac.uk/ena, accessed on 1 July 2020) database by means of the newly implemented meta-search engine interface using the search keywords ‘soil metagenome’, ‘root metagenome’, ‘ILLUMINA’, and ‘whole-genome-shotgun (WGS) metagenome’, thus trying to exclude 16S rRNA gene amplicon sequencing projects. Due to the ambiguity of the description in the ENA database, samples matching two or more environments (e.g., matching both keys words ‘root metagenome’ and ‘rhizosphere metagenome’) were manually checked and corrected. Particular soil metagenome datasets from European soil habitats of interest are included in the downloaded set of projects. Fragment recruitments using the genomes of the selected strains as templates were performed by application of the bioinformatics tool SparkHit [30]. Corresponding computations were scaled-up and parallelized by using the de.NBI Cloud environment (https://www.denbi.de/cloud, accessed on 1 July 2020). We implemented a fast and sensitive fragment recruitment tool, called Sparkhit-recruiter. Sparkhit-recruiter extends the Fr-hit [31] pipeline, and is implemented natively on top of the Apache Spark and integrates a series of analytical tools and methods for various genomic applications. The fragment recruitment option implements the q-Gram algorithm to allow more mismatches than a regular read mapping during the alignment, so that extra information is provided for the metagenomic analysis. Finally, we applied SparkHit on all soil metagenome FASTQ files as available at UNIBI’s de.NBI Cloud object storage. The first 10 million reads of each FASTQ file were compared to all 20 PGP reference genomes selected within this study. The alignment identity threshold was set to >97% to only identify closely related genomes.

To remove highly covered regions on the genome (Appendix A) that could be introduced by homologous genes of other microbial genomes or 16S rRNA genes that are repetitive in the genomes, a peak removal step was applied to pre-process the fragment recruitment results. The mean coverage of the genome was calculated, as well as the standard deviation (SD) of all the read coverages on the genome. Recruited reads that are located at those loci with coverages of more than the mean value plus two times of the standard deviation value (Algorithm 1) are removed, as their coverages are abnormally high.
**Algorithm 1:** Filter Recruited Reads that Are Located at Highly Covered Regions**Input:** recruited reads *R***Output:** qualified reads *Q*1 Allocate and initialize array of all recruited reads2 **for**
*i = 1, 2, …, R*
**do**3  *l* = locus of each read *R_i_*4  *D(l)* = dictionary of coverage at each locus *l*5  *D(l)*++6 **end**7 *M* = MeanValue(*D*)8 *SD* = StandardDeviation(*D*)9 **for**
*l = 1, 2,.., D*
**do**10 F = dictionary of unqualified loci on the genome11 **if** D(l) <= M + 2*SD **then**12  F(l) = true13 **else**14  F(l) = false15 **end**16 **for**
*i = 1, 2, …, R*
**do**17  *l* = locus of each read *R_i_*18  **if**
*F(l)* == true **then**19   add *R_i_* to *Q*20  **end**21 **end**

To measure the abundance of the PGP genomes along all soil, root, and rhizosphere metagenome samples, the fragment recruitment counts are normalized by the total read number of each sample. In total, 3230 million reads from soil metagenome datasets, 1400 million reads from root metagenome dataset, and 4500 million reads from rhizosphere metagenome datasets were used for the fragment recruitment. The fragment recruitment counts of each sample were then normalized to a total of 1000 million reads using the following formula:$$N=R\times \frac{1,000,000,000}{T},$$
where *N* denotes the normalized fragment recruitment counts, *R* represents the read number that are recruited to the genome, and *T* denotes the total reads in a given metagenome sample.

### 2.3. Microbe-Microbe In Vitro Compatibility Test

The design, formulation, and optimization of effective microbial consortia as inoculants require evidence of the ability of the consortium members to coexist. Therefore, microbial strains (22 bacterial strains and the yeast *K. pastoris* PP59; see Section 3.3) were subjected to in vitro compatibility test applying the agar diffusion test as described by Irabor and Mmbaga [32], with minor modifications.

A loop of each microbial strain was picked up from glycerol stocks stored at −80 °C and streaked onto nutrient agar (NA) plates. After microbial growth at 28 °C for 24–48 h, 3–4 isolated colonies were transferred to 4 mL of Nutrient Broth (NB) medium and incubated overnight at 28 °C and 200 rpm. One hundred microliters of the test microorganism of about 10^8^–10^9^ colony forming units per mL (CFU/mL) were spread on the surface of fresh NA plates. Sterilized filter paper discs (5-mm diameter, Whatman number 1) were placed on the spread plate (maximum five discs/plate), and each of them was inoculated with 10 μL of an overnight NB-grown culture of the microorganism (as stated above) to be tested against to check compatibility. Plates were incubated at 28 °C and observed at 24-h intervals over a period of 4 days. Two microorganisms were considered *compatible* as they were capable to grow together showing overlap in the area beyond the disc border. On the other hand, they were identified as *incompatible* in the cases in which a clear zone of inhibition was observed around the disc. When neither the inhibition zone nor the overgrowth around the disc was observed, the compatibility was considered *unclear*. For each bacteria-bacteria or bacteria-yeast combination, two independent experiments were performed with three replicates.

The presence of fungal strains among the selected PGPMs (i.e., *T. harzianum* TH01 and *T. harzianum* ATCC^®^ 48131^TM^; see Section 3.3) required the additional investigation of the in vitro bacteria-fungi compatibility. The agar plate method as described by Siddiqui and Shaukat [33] was adopted with minor modifications. A loop of each microbial strain (bacteria and/or yeast) was picked up from glycerol stocks stored at −80 ° C and streaked onto nutrient agar (NA) plates. The plates were incubated at 28 °C for 24–48 h to allow the microbial growth. In the second step, a loop of each bacterium or yeast to be tested (maximum four strains for each plate) was streaked near the edge of potato dextrose agar (PDA) plates at fixed positions. A mycelium agar plug (5-mm diameter) of the fungus (9 days old) was transferred to the centre of the previous inoculated PDA plates. PDA plates inoculated only with the fungus served as control. The plates were incubated in darkness at 28 °C, and the zone of inhibition (if any) was recorded after 48–96 h of microbial growth. Any overgrowth of the fungus on microbial (bacteria and/or yeast) streaks without a zone of inhibition were considered as bacteria-fungus and yeast-fungus compatibility. For each microbial combination (bacteria-fungus and yeast-fungus), two independent experiments were performed with three replicates.

### 2.4. Effects of Bioactive Compounds on Microbial Growth In Vitro 

The following bioactive compounds were tested: (i) agro-industrial sugar cane molasses, rich in humic and fulvic substances, free amino acids, peptides and glycine betaine (BS1); (ii) protein hydrolysates obtained by autolysis of previously grown *Saccharomyces cerevisiae* yeast, rich in high free amino acids, peptides, nucleotides, B vitamins, trace elements, and other growth factors (BS2); (iii) animal protein hydrolysates derived from cheese whey (BS3); and (iv) agro-industrial brewery by-products residues (BS4). In vitro tests were performed to evaluate the prebiotic and biostimulant activity of bioactive compounds BS1–BS4 (AGRIGES srl, Italy). Strains composing the microbial consortia (MC_A, MC_B, and MC_C) were grown in starvation conditions, with different concentrations of the bioactive compounds. Briefly, the bioactive compounds were dissolved in water, filtered (0.22 μm Ø size, Millipore), sterilized and included in water agar (WA) plates at different concentrations ranging from 10 ppm to 10,000 ppm. Microorganisms (bacteria and/or yeast) were grown as described above (see Section 2.3). One microliter of overnight bacterial or yeast suspension was streaked on WA plates (1.5% *w*/*w*) with and without the bioactive compounds, and on NA (positive control). Microbial growth was examined after 24- and 48-h of incubation at 28 °C and compared with positive and negative control plates (NA and WA without bioactive compounds, respectively). The fungal strain *T. harzianum* TH01 was tested in a separate assay. A 5-mm block of 5 days old pure culture of fungal strain was placed upside down at the centre of WA plates containing different concentrations of bioactive compounds (from 10 ppm to 10,000 ppm) and PDA plates (positive control). Subsequently, the plates were incubated at 28 °C for 72 h. The fungal radial growth (cm) was recorded at right angles of agar plates by the aid of a ruler and compared with positive and negative control plates (PDA and WA without bioactive compounds, respectively).

Statistical data analysis was performed using the open source program R (version 4.0.2) with RStudio (R Core Team, Vienna, Austria, version 1.2.5033). A fitting linear model was developed to analyze the fungal radial growth. For data comparison, a two-way repeated measures ANOVA was conducted. To identify significant differences between the means of different treatments, a Student-Newman-Keuls (SNK) test was performed.

## 3. Results and Discussion

### 3.1. Identification of the Most Promising Beneficial Microorganisms and Carriers 

The choice of the PGPMs is fundamental to develop efficient synthetic microbial consortia capable to promote the growth and health of crop plants [11]. To identify the most promising PGPMs for maize, wheat, tomato and potato plants, an extensive literature survey was carried out. According to our inclusion criteria, a total of 134 published articles were retrieved and provided to be eligible to identify the most promising PGPMs for each crop. The main findings of all collected manuscripts are reported in Data Sheet S1. The literature survey showed that several PGPMs can be used effectively to promote plant growth in normal and stressful environments; however, their real effectiveness under field conditions could hardly be evaluated due to the high variability in the efficacy and reproducibility in several environmental conditions. Appendix A summarizes the database results with respect to the eligible articles, as well as the number of studies that were positively considered, including the number of PGPMs species and commercial products. The list of PGPMs and commercial biofertilizers for each crop (tomato, maize, potato, and wheat) deriving from literature survey is reported in Appendix A. Results showed that a great variety of microorganisms belonging to different genera and species were found to improve the growth of the four crops, with microorganisms belonging to the *Bacillus* and *Pseudomonas* genera appearing the most frequently considered. Moreover, also several species of the *Streptomyces* and *Trichoderma* genera were tested as bioinoculants for wheat. Several commercial biofertilizers have been developed and tested for tomato, maize, and wheat, whereas only two commercial biofertilizers have been developed for potato. A few commercial biofertilizers were applied on more than one of the four crops of interest. Following the evaluation process, scientific articles reaching the score of 10 points, permitted us to identify PGPMs as potential candidates for MC set-up. The list of PGPMs selected for the study is shown in Table 1. Both strains deriving from literature survey and from internal microbial collections are represented.

The effects of PGPMs can be exerted if there is an effective delivery system to bring the microorganisms near the roots. For this purpose, a literature search was also carried out to identify the most recent evidence on the use of carriers to deliver microbial inoculants, and trends in agricultural applications. A list of potential carriers for delivering microbial consortia to crop plants is reported in Appendix A. Among the characteristics that help identifying a suitable carrier, the following have to be considered: low cost, good availability, adequate shelf life of the product, easiness in distribution to and within the soil, good moisture absorption capacity, easiness in sterilization, good pH buffering, chemical and physical stability, biodegradability, non-polluting properties, and environmental safety [62]. No perfect carrier having all the mentioned criteria exists, but the delivery methods chosen should possibly have most of them. Appendix A reports advantages and disadvantages of different carriers, along with successful examples of applicability. Based on the type of carrier, distribution systems can be delivered on seeds [63], roots, or plants in the field [64]. In addition to carrier-based formulations, liquid suspensions, or water-in-oil emulsions of microbial cells, spores or conidia can be considered [65].

### 3.2. Metagenome Fragment Recruitments on Genomes of Candidate PGP Soil Microorganisms Represented in Databases

To evaluate natural occurrence of the selected PGPMs in different soil and rhizosphere environments, large-scale metagenome fragment recruitments were accomplished. Obtained results provided insights into adaptation properties of PGPMs to specific soil/rhizosphere types and conditions. Corresponding information can help to identify the most suitable and promising PGPM for a specific target soil habitat and associated conditions.

Genomes of PGP strains featuring the highest probabilities to be similar to genomes represented within the selected soil metagenomes (>97% identity of individual reads versus the selected reference genomes) are recorded as outcome of the fragment recruitment approach. Likewise, information on the origin and characteristics of the soil metagenomes harboring PGP strains of interest can be extracted from stored metadata associated with the identified metagenome datasets. Details on the applied methodology for fragment recruitments are described in the Materials and Methods Section 2.2.

Results of the soil metagenome fragment recruitments showed that the genome of the PGP strain *Bacillus subtilis* subsp. *subtilis* str. 168 is represented in soil metagenomes deposited in the ENA SRA by the Nanjing Agriculture University without any further meaningful metadata (PRJNA343989) and in bulk soil microbial communities from a forest located near Harvard (USA; PRJNA365880). In these cases, respectively, 0.12% and 0.1% of the soil metagenome sequence reads mapped to the genome of the target PGP strain. Likewise, further PGP species, such as *Pseudomonas fluorescens*, *Burkholderia ambifaria*, and *Stenotrophomonas rhizophila* were identified in different datasets referring to, e.g., *Arabidopsis*, *Brassica*, *Sorghum*, *Miscanthus*, and corn rhizosphere samples. A summary overview on the obtained results is shown in Figure 1. PGP strains were also identified in metagenomes from soil enrichment cultures and isolated microbial consortia that do not represent native soil microbiomes and, therefore, will not be considered any further in this analysis. It was observed that there are relatively few sequence reads featuring perfect matches to the reference genomes. These results indicate that the genomes of the selected PGP strains are related to homologous genomes of the analyzed soil metagenomes but are not identical.

Fragment recruitments were also carried out for metagenome datasets obtained for rhizosphere and root microbiomes since many PGP microbial species are rhizosphere competent. In comparison, most PGP species have higher fragment recruitment abundances in root and rhizosphere samples than in soil samples (see Figure 1). The top ten list of reference genomes receiving the most recruited metagenome reads includes *Pseudomonas putida*, *Bacillus simplex*, *Stenotrophomonas rhizosphila*, *Bacillus megaterium*, *Raoultella terrigena*, and *Pseudomonas fluorescens* with 5.97 to 1.4% of the metagenome sequence reads matching to the genomes of these species (Table 2). Corresponding metagenome datasets represent citrus, pomegranate, *Sorghum*, switchgrass, and *Arabidopsis* rhizosphere samples. Many more reference genomes were identified in other rhizosphere metagenomes but with lower abundances. For example, *Pseudomonas fluorescens* F113 received 0.28 % of the metagenome reads from maize rhizosphere samples. However, since it was present in most of the rhizosphere samples, it still has a high total abundance in the rhizosphere environment (Figure 1A). At a cut-off threshold of 0.01 % matching sequence reads, neither metagenomes from tomato nor potato rhizosphere samples were identified.

Regarding the large-scale fragment recruitment use case, it can be concluded that genomes of the selected PGP strains are not very well represented in most of the tested bulk soil microbiomes suggesting that other PGP bacteria are better adapted to and are more competitive in these soil habitats. To acquire genome sequence information of potentially new PGP bacteria that are better adapted to the habitats analyzed, metagenome assembly and binning approaches have to be applied to yield Metagenomically Assembled Genomes (MAGs). These will provide the basis for further characterization of putative novel PGP soil microbiome members including reconstruction of their metabolism and lifestyle. However, the tested PGP strains seem to be better adapted to the root and rhizosphere of particular plants which should be considered for the design of application formulations and procedures.

### 3.3. Evaluation of In Vitro Co-Culture Compatibility of Selected Microbial Strains

The selection of suitable and compatible strains is one of the prerequisites in the use of multi-strain inoculants and represents a crucial aspect in formulating synthetic microbial consortia as bioinoculants [3,79]. Following the results of the literature survey and taking also into account the microbial strains with pre-established growth supporting abilities available from SIMBA consortium (Table 1), a total of 25 microbial strains were selected (see Table 1). Twenty-two out of 25 are bacteria belonging to the genera *Azospirillum*, *Azotobacter*, *Agrobacterium*, *Bacillus*, *Burkholderia*, *Komatagaella*, *Paraburkholderia*, *Ranhella*, and *Raoultella*, one is a yeast, belonging to the genus *Komatagaella*, and two are fungi, belonging to the genus *Trichoderma*. For bacteria-bacteria and bacteria-yeast compatibility, Nutrient Agar (NA) was employed as it is a wide-spectrum medium and because all the strains were able to grow on it (data not shown). The results of in vitro compatibility of selected bacterial strains and the yeast *K. pastoris* are reported in Table 3.

Among the 23 examined microbial strains (22 bacteria and one yeast), most were compatible with each other and thus could coexist (Table 3 and Figure 2). This in turn, allows them to be a part of the specific microbial mixtures. Few incompatibilities were observed, mainly involving the following strains: *Bacillus* sp. BV84, *B. amyloliquefaciens* BA41, *B. licheniformis* PS141, *B. subtilis* LMG 23370 and LMG 24418 strains, *B. amyloliquefaciens* LMG 24415 and LMG 9814 strains, and, to a lesser extent *A. brasilense* ATCC 29710, *P. fluorescens* PN53 and *B. ambifaria* LMG 11351. In addition, *A. chroococcum* DSM 2286 resulted incompatible with most tested bacteria. Unclear compatibility of *P. tropica* MDIIIAzo225 was observed with *B. amyloliquefaciens* LMG 9814 and *B. subtilis* LMG 23370, *P. fluorescens* DR54 and, *P. fluorescens* PN53 and *K*. *pastoris* PP59, and *B. ambifaria* MCI 7. Likewise, unclear compatibility of *B. amyloliquefaciens* LMG 9814 with *A. brasilense* ATCC 29710 was observed. Thus, those microorganisms being incompatible with others or showing an unclear compatibility were not considered in further experiments. Their high sensitivity or strong inhibitory effects make them incompatible and unable to work together in a microbial consortium.

Based on the above explained compatibility tests, 22 bacterial strains and the yeast *K. pastoris* were further tested for their in vitro compatibility with *T. harzianum* strains. *T. harzianum* strains TH01 and ATCC 48131, previously identified (Table 1) were considered and tested in the in vitro assay for bacteria-fungi and yeast-fungi compatibility. The absence of an inhibition zone around the disk indicated that microbial strains were compatible with *T. harzianum* (Figure 2). Results of the assays revealed that 12 out of 23 bacteria were compatible with *T. harzianum* TH01, whereas only 7 out of 23 bacteria were compatible with *T. harzianum* ATCC 48131. The yeast *K. pastoris* PP59 resulted compatible with both *T. harzianum* strains (Table 4).

### 3.4. Design of Microbial Consortia Inoculants

The choice of the appropriate inoculum represents a key step towards the development of a successful biofertilizer [80]. Multi-strain PGPMs mixtures, so called microbial consortia, appear to have greater efficacy on improvement of plant-growth than single strains [81]. There are two main strategies to select consortium members: (1) top-down or natural microbial consortia (from complex to simple) approach: the consortium members are the identified keystone players from one specific complex microbial community [18] and (2) the bottom-up or synthetic microbial consortia (from simple to complex) approach: the consortium members are selected from an extensive pool of microorganisms isolated from various sources, which may possess the desired traits [79,82,83]. In the present work, a bottom-up approach was used to design simple microbial communities with a defined composition of 5 or more species/strains. According to the results of bacteria-bacteria and bacteria-fungi compatibility tests (Table 3 and Table 4), several microbial isolates belonging to different genera and species could be chosen to design suitable microbial consortia inoculants. The use of diverse microorganisms that can promote plant growth and protect plants from biotic or abiotic stresses with different modes of action is considered a basic requirement in the engineering of synthetic microbial mixtures applied to plants [15,84]. Here, the following selection criteria were used to formulate the microbial consortia: ability to coexist, different functionalities and PGP activities as reported by literature. Three microbial consortia inoculants named MC_A, MC _B, and MC _C were developed in which a specific function (i.e., nitrogen fixation, P-solubilization, biocontrol, amylolytic activity, auxin, or auxin-like compounds production) was represented by at least one member. Microbial consortium A comprises six microorganisms, among which four are bacteria (*A. chrococcum* LS132, *B. licheniformis* PS141, *P. tropica* MDIIIAzo225, *P. granadensis* A23/T3c), one is a yeast (*K. pastoris* PP59), and one is a fungus (*T. harzianum* TH01), whereas consortium B and C are each composed by five bacteria alone. Microbial consortium B consists of *A. vinelandii* DSM 2289, *Bacillus* sp. BV84, *B. amyloliquefaciens* LMG 9814, *P. fluorescens* DR54, *R. aquatilis* BB23/T4d, while microbial consortium C is composed of *A. chroococcum* LS132, *B. amyloliquefaciens* LMG 9814, *B. ambifaria* MCI 7, *R. aquatilis* BB23/T4d and *P. fluorescens* DR54. Most of these microorganisms have previously been proven to exert a PGP effect on plant growth (for more detail, see Table 2). In particular, *T*. *harzianum* TH01, *K*. *pastoris* PP59, and *Bacillus* sp. BV84 are already present in the commercial product Micosat F (CCS Aosta, S.r.l, Italy). The PGP and biocontrol properties of *T. harzianum* strains are well recognized [85]. The application of *T. harzianum* may favour the biocontrol of several fungal pathogens, the promotion of plant growth and the increase of nutrient availability and drought stress resistance [42,86,87,88]. Concerning the other strains, *P. granadensis* A23/T3c (formerly *P. fluorescens* A23/T3c) was reported to increase the growth of *Sorghum bicolor* in greenhouse, and *R. aquatilis* BB23/T4d (formerly *Enterobacter* sp. BB23/T4d) showed a positive effect on root growth of *Sorghum bicolor* alone and in dual strain inocula with *Burkholderia ambifaria* (formerly *Burkholderia cepacia*) PHP7 [54]. *B. ambifaria* MCI 7 (formerly *Burkholderia cepacia* MCI 7) is a promising biopesticide and plant-growth-promoting inoculant for maize being able to determine an increased growth response of maize by a direct and indirect mechanism, in both uninfested and infested soil with *Fusarium moniliforme*, under greenhouse conditions [52,53,89], while *P. fluorescens* DR54 was found to inhibit the growth of the root pathogen *Phytium ultimum* in the sugar beet rhizosphere in pot experiments [55] and to increase the highly soluble phosphorous in soil after its application on maize under field conditions [52]. *Paraburkholderia tropica* MDIIAzo225 is a nitrogen fixing bacterium isolated from maize rhizosphere (unpublished results ENEA). Strains belonging to this species have been proven to exert PGP on several crops to do their ability to form biofilm and colonize plant-tissues [90,91]. Among the two strains belonging to the *Azotobacter*, *A. vinelandii* DSM 2289 was reported to increase the growth, chlorophyll content and iron content of soybean plants grown in calcareous soil [40], whereas *A. chroococcum* LS132 is a nitrogen fixing bacterium with the ability to improve the growth of tomato plants (unpublished results, AGRIGES). *B. amyloliquefaciens* LMG 9814 is a PGP strain known to produce alpha-amylase, alpha-glucosidase, iso-amylase production, and *B. licheniformis* PS 141 is an indole acetic acid (IAA) producing bacterium isolated from rhizosphere (unpublished results, AGRIGES).

The selected microbial consortia inoculants have the potential to endow the plant with significant growth enhancement due to various hormones and other metabolites contributed by each participating member of the consortia acting in synergistic way. Microbial consortia have the potential to establish novel microbial communities in the rhizosphere and may result in new PGP effects not obtained by using single inoculants [3].

### 3.5. Prebiotic Effect of the Bioactive Compounds

Bioactive compounds are widely used as plant biostimulants, for their ability to increase crop productivity and ameliorate crop tolerance to abiotic stresses [27,28,29]. Some recent studies have described the beneficial effects of bioactive compounds like humic acids combined with beneficial microbes [92,93], providing evidence that they can promote growth of beneficial soil microorganisms. Here, we aimed to identify the suitable bioactive compounds to be used in combination with microbial consortia to rapidly increase the number of beneficial microorganisms in the soil and activate soil nutrients, acting as prebiotic and biostimulant activity. We hypothesize that the effect of the combined application of microbial consortia inoculants and bioactive compounds into the soil can effectively improve crop yield and quality.

In the present study, to identify the more suitable and functional bioactive compounds capable to support the growth of the microorganisms composing the selected microbial consortia, a set of organic protein hydrolysate compounds were tested for their compatibility with the single microbial strains belonging to MC_A, MC_B, and MC_C microbial consortia. To achieve this goal, each single PGPM composing the identified consortia was grown in starvation conditions with several concentrations of bioactive compounds. Overall, results revealed that the bioactive compound BS2 exerted a positive effect on the growth of all microbial strains tested in starvation conditions, BS1 showed a positive effect on 8 bacterial strains, whereas BS3 showed a prebiotic effect towards *T. harzianum* TH01 only (Table 5). BS4 was not able to induce reproducible effects, probably due to the intrinsic high variability of its components.

In Figure 3, an example of results obtained by using BS2 compounds on bacterial growth after 48-h incubation is shown. As it can be seen from the qualitative assay, the addition of at least 1000 ppm fostered the bacterial growth in starvation conditions.

Concerning the effect on *T. harzianum* TH01, the bioactive compounds BS1, BS2, and BS3 resulted to positively affect the fungal growth (Table 5). Figure 4 shows the radial increments of *T. harzianum* TH01 grown in starvation condition (WA) in the presence of increasing concentrations of the bioactive compound BS2 at 48 and 72 h.

Quantitative results obtained by measuring the radial growth of *T. harzianum* TH01 over time (24–48–72 h) are shown in Appendix A. However, the data confirmed a positive effect only for BS2 and BS3. Regarding BS1, a negative impact on fungal growth was observed, especially after 48 and 72 h, suggesting a partly growth inhibitory effect of BS1. Comparatively strong growth reductions were observed if BS1 was added at 10,000 ppm. In contrast to BS1, the addition of BS2 and BS3, at 10, 100, 1000, and 10,000 ppm fostered a significant increase of fungal growth in WA plates at 24, 48 and 72 h (*p* < 0.05) (Appendix A, and Figure 5). After a growing period of 72 h, *T. harzianum* TH01, cultivated on WA plates supplemented with BS2 at 10,000 ppm, achieved the same radial growth as on PDA plates. A similar evidence of a strong fungal growth promotion was also observed for BS3 after 48 and 72 h. The fungal growth edge achieved after 48 h upon the application of BS3 at 10,000 ppm was significantly greater in comparison to all other treatments. Hence, using this concentration, a notable growth increase of *T. harzianum* TH01 of almost 20% was detected compared to the growth rate of the fungus on PDA plates. In addition, and regardless of the bioactive compound concentration applied to WA plates, the growth of *T. harzianum* TH01 was significantly increased compared to the negative control (WA without addition of BS) in all plates after 72 h. The herein observed growth promotional effect of BS3 at any concentration even allowed the fungus to achieve the same order of magnitude in radial growth as it was observed on PDA plates. Based on the results obtained in this study, a strong growth promotional effect of different bioactive compounds on different PGPMs can be expected.

Overall, in vitro results suggest the presence of bioactive compound BS2 and BS3 fostered a rapid increase of microbial growth in starvation condition. The combination BS-microbial consortia could represent a valid strategy for the development of new biostimulants for a sustainable agriculture.

## 4. Conclusions

In the recent years, the need to define and adopt more sustainable and environmentally friendly agriculture practices has been well recognized. The coronavirus disease (COVID-19) pandemic has led to increasing doubts about possible impacts of intensive, non-sustainable agriculture on the general equilibrium of man, animals, and nature [12]. The interest for environmental-friendly solutions in modern agriculture results from the trend to search for natural strategies that reduce the application of chemicals in agriculture. Today, it is accepted that microorganisms thrive in diverse natural environments in complex microbial communities. Hence, the use of beneficial microorganisms combined in consortia is very promising for improving crop yield and quality representing a reliable and eco-friendly solution that may respond to the challenges for modern agriculture. A well-designed application of natural microorganisms and organic amendments can greatly increase the plant yield or the control of plant pathogens in an environmentally sustainable way. In the present study, a bottom-up approach was taken into-account to identify microbial consortia for sustainable agriculture. Three MC, composed by compatible multi-strain species, were identified by synthetic assemblages of isolated PGPMs with different functions following by the in vitro analysis of their ability to coexist and in vitro test with bioactive compounds. The findings presented in this study indicate that bio-active compounds can enhance the growth of beneficial microorganisms composing the selected MC, suggesting that signals produced by these PGPMs can act synergistically with the organic compounds to enhance plant growth and productivity. Exploiting the efficacy in greenhouse and field trials of the combined MC-bioactive compounds, as well as their overall ecological impact on the native soil microbiome, will permit to define new plant biostimulants for a more sustainable and resilient agriculture.

## Figures and Tables

**Figure 1 microorganisms-09-00426-f001:**
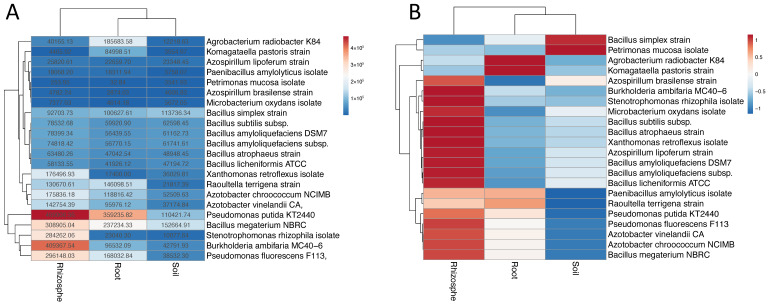
Heatmaps of fragment recruitment counts on 22 plant growth promoting (PGP) microbial genomes in three different environments (Rhizosphere, Root, and Soil according to descriptions provided in the metadata of the analyzed datasets). (**A**) The heatmap was plotted using normalized fragment recruitment counts (normalized to the total number of sequence reads of each environment). Both, rows and columns, are clustered using the Euclidean distance and average linkage method. (**B**) Rows of the heatmap are scaled using unit variance scaling (Autoscaling). Both, rows and columns, are clustered using the Euclidean distance and average linkage method. Please note that the different order of species designations between panels A and B is due to the applied cluster algorithm.

**Figure 2 microorganisms-09-00426-f002:**
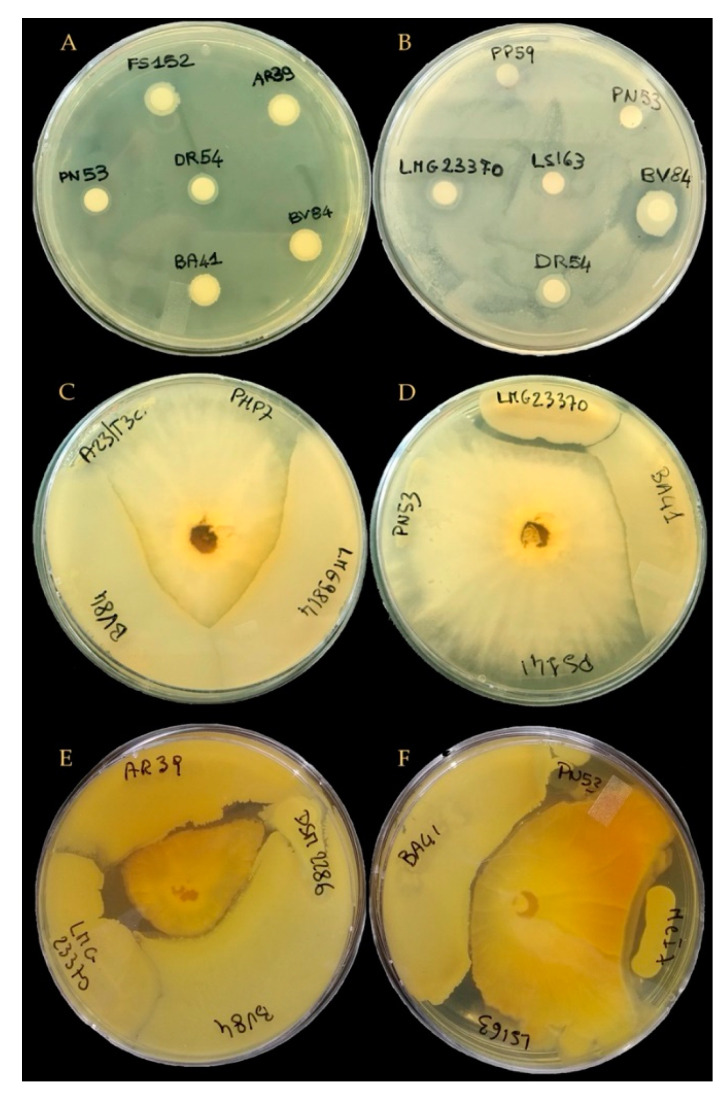
Examples of the in vitro microbial interactions. (**A**) Test strain *B. ambifaria* MCI 7: compatible with *B. amyloliquefaciens* BA41, *A. radiobacter* AR39 and *B. licheniformis* FS152, unclear compatibility with *Pseudomonas* sp. PN53 and *P. fluorescens* DR54; (**B**) test strain *B. licheniformis* PS141: compatible with *K. pastoris* PP59, *Pseudomonas* sp. PN53, *A. chroococcum* LS163, and *B. subtilis* LMG 23370, incompatible with *Bacillus* sp. BV84; (**C**) test strain *T. harzianum* TH01: compatible with *P. granadensis* A23/T3c and *B. ambifaria* PHP7, incompatible with *Bacillus* sp. BV84 and *B. amyloliquefaciens* LMG 9814; (**D**) test strain: *T. harzianum* TH01: compatible with *P. fluorescens* PN53 and *B. licheniformis* PS141, incompatible with *B. subtilis* LMG 23370 and *B. amyloliquefaciens* BA41; (**E**,**F**) test strain: *T. harzianum* ATCC 48131: incompatible with *A. radiobacter* AR39, *A. chroococcum* DSM 2286, *B. subtilis* LMG 23370, *Bacillus* sp. BV84, *B. amyloliquefaciens* BA41, *B. ambifaria* MCI 7, *Pseudomonas* sp. PN53, and *A. chroococcum* LS163.

**Figure 3 microorganisms-09-00426-f003:**
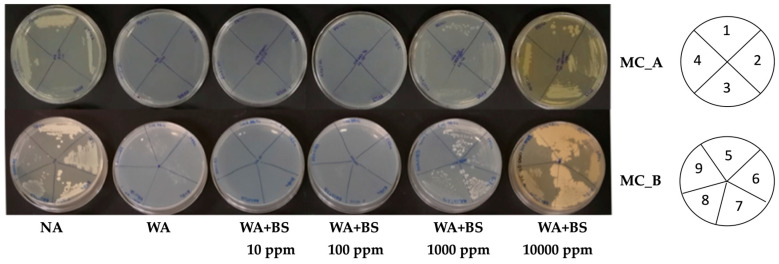
Bacterial strains belonging to microbial consortia (MC)_A (on the top) and MC_B (on the bottom) growing in starvation conditions in the presence or absence of BS2 ranging from 10 ppm to 10,000 ppm. Tested strains for MC_A: *B. licheniformis* PS141 (**1**), *A. chroococcum* LS132 (**2**), *K. pastoris* PP59 (**3**), *P. granadensis* A23/T3c (**4**), MC_B: *B. amyloliquefaciens* LMG 9814 (**5**), *P. fluorescens* DR54 (**6**), *Bacillus* sp. BV84 (**7**), *R. aquatilis* BB23/T4d (**8**), and *A. vinelandii* DSM 2289 (**9**). For a better view of the position of each strain, a plating scheme was added to the right. The plates on the right report the numbers corresponding to each of the tested strains in MC_A (**1**–**4**) and MC_B (**5**–**9**). The bio-assay was performed in triplicate. NA = Nutrient agar; WA = water agar; BS = Biostimulant.

**Figure 4 microorganisms-09-00426-f004:**
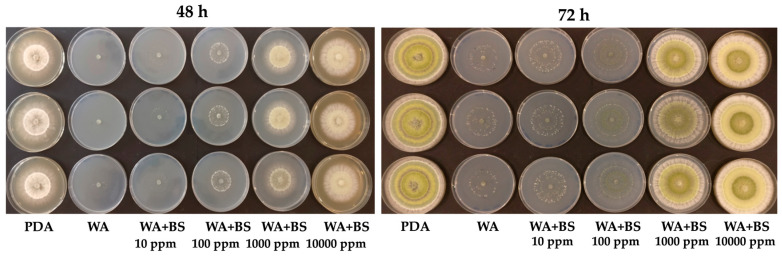
*T. harzianum* TH01 growing in starvation conditions in the presence or absence of BS2 ranging from 10 ppm to 10,000 ppm. On the left, the effects of BS2 at 48 h, on the right the effects at 72 h. The bio-assay was performed in triplicate. PDA = Potato Dextrose Agar; WA = water agar; BS = Biostimulant.

**Figure 5 microorganisms-09-00426-f005:**
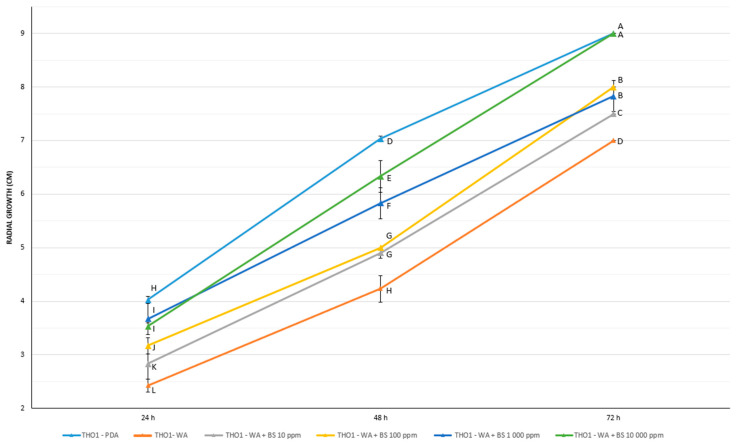
Prebiotic effect of BS2 on *T. harzianum* TH01 growth in starvation conditions. WA = water agar; PDA = Potato Dextrose Agar; BS2 = Bioactive compounds BS2. A two-way repeated ANOVA with Student-Newman-Keuls (SNK) test was performed for data analysis. Different capital letters (A–L) indicate significant differences between treatments (*p* < 0.05).

**Table 1 microorganisms-09-00426-t001:** List of the Strains Which Were Selected as Potential Plant Growth-Promoting Microorganisms (PGPMs) in this Study, Their Isolation Source, and Properties. The Species Names Were Verified According to the Latest Classification in the NCBI Taxonomy Database.

PGPMs	Strain	Origin	Country	Properties	References
*Acaulospora morrowiae*	CL290	Rhizosphere	USA	PGP	[34]
***Agrobacterium radiobacter ****	**AR 39**	Soil near peach tree	Italy	Biocontrol/PGP	Unpublished results
***Azospirillum brasilense ****	**ATTC 29710**	*Cynodon dactylon* rhizosphere	USA	N-fixation	[35]
***Azospirillum brasilense ****	**NCCB 78036**	Soil under soy field	India	N-fixation	Unpublished results
*Azospirillum lipoferum*	CRT1	Field grown maize	France	N-fixation	[36]
*Azotobacter chroococcum*	76A	Soil	Italy	N-fixation	[37]
***Azotobacter chroococcum ****	**DSM 2286**	Unknown	unknown	N-fixation	[38,39]
***Azotobacter chroococcum ****	**LS132**	Rhizosphere	Italy	N-fixation	Unpublished results
***Azotobacter chroococcum ****	**LS163**	Rhizosphere	Italy	N-fixation	Unpublished results
*Azotobacter chroococcum*	S-5	Unknown	Iran	N-fixation	[38,39]
***Azotobacter vinelandii ****	**DSM 2289**	Unknown	unknown	Siderophore production, N-fixation	[40,41]
***Bacillus* sp.**	**BV84**	Grape leafs	Italy	Biocontrol/PGP	Unpublished results
***Bacillus amyloliquefaciens ****	**BA41**	Wheat rhizosphere	Italy	Biocontrol/PGP	Unpublished results
*Bacillus amyloliquefaciens*	FZB42	Plant pathogen infested soil	Germany	Biocontrol/PGP	[42]
***Bacillus amyloliquefaciens ****	**LMG 9814**	Soil	UK	Alpha-amylase, alpha-glucosidase, iso-amylase production	Unpublished results
*Bacillus atrophaeus*	ABI02A	NA	Germany	PGP	[43]
***Bacillus licheniformis ****	**PS141**	Rhizosphere	Italy	Indole acetic acid (IAA) production	Unpublished results
*Bacillus megaterium*	M3	Rice	unknown	P-solubilization	[44,45]
*Bacillus megaterium*	PMC 1855	Unknown	unknown	P-solubilization	[46]
***Bacillus pumilus ****	**LMG 24415**	Soil	Ecuador	PGP	[47]
*Bacillus simple*	R49538	Unknown	Ecuador	PGP/IAA production	[47]
*Bacillus subtilis*	FZB24 WG	NA	Germany	Biocontrol/PGP	[48,49]
***Bacillus subtilis ****	**LMG 23370**	Forest soil	India	Biocontrol/PGP	Unpublished results
***Bacillus subtilis ****	**LMG 24418**	Soil	Ecuador	PGP	[47]
*Bacillus subtilis*	OSU-142	pepper	unknown	N-fixation, biocontrol	[50,51]
***Burkholderia ambifaria ****	**MCI 7**	Maize rhizosphere	Italy	PGP	[52,53]
***Burkholderia ambifaria ****	**PHP7/LMG 11351**	Maize rhizosphere	France	PGP	[54]
*Gigaspora gigantea*	PA125	Rhizosphere	USA	PGP	[34]
*Gigaspora rosea*	NY328A	Rhizosphere	USA	PGP	[34]
***Komagataella pastoris ****	**PP59**	Grape rhizosphere	Italy	PGP	Unpublished results
*Paenibacillus* sp	R47065	Unknown	Ecuador	PGP/IAA production	[47]
***Paraburkholderia tropica***	**MDIIIAzo225**	Maize rhizosphere	Italy	N-fixation	Unpublished results
***Pseudomonas granadensis *****	**A23/T3c**	Soil	Italy	PGP	[54]
***Pseudomonas fluorescens ****	**DR54**	Sugar beet rhizosphere	Denmark	Biocontrol	[55]
*Pseudomonas putid*	P1-20/08	Soil	Ecuador	PGP	[47]
***Pseudomonas* sp. ***	**PN53**	Grass rhizosphere	Italy	PGP	Unpublished results
***Rahnella aquatilis *****	**BB23/T4d**	soil	Italy	PGP	[54]
***Raoultella terrigena ****	**FS152**	Rhizosphere	Italy	Phytase activity, siderophore production	Unpublished results
*Rhizophagus intraradices* ***^§^***	FR121 ^§^	-	-	Tolerance to abiotic /biotic stress	[56,57]
*Septoglomus constrictum*	FL328	Rhizosphere	USA	PGP	Unpublished results
*Streptomyces* sp.	SA 51	Rhizosphere	Italy	Biocontrol	Unpublished results
*Trichoderma gamsii*	6085	uncultivated soil	Crimea (UA)	Biocontrol	[58]
*Trichoderma harzianum*	OmG-08	Orchid roots	Germany	P-solubilization	[59]
*Trichoderma harzianum*	OmG-16	NA	Germany	P-solubilization	[49]
*Trichoderma harzianum*	T6776	Soil	Italy	Biocontrol/PGP	[60]
***Trichoderma harzianum ****	**TH01**	Grass soil and rhizosphere	Italy	PGP	Unpublished results
***Trichoderma harzianum ****	**CBS 354.33/** **ATCC 48131**	Soil	USA	Chitinase production, biocontrol	[61]

In bold (both single and double-asterisks) strains analyzed in the in vitro assay. ** Double-asterisks denote the new taxonomic assignment of two PGPMs not previously identified at species level. *Pseudomonas* sp. A23/T3c strain and *Enterobacter* sp. BB23/T4d strain, respectively, were subjected to 16S rDNA sequencing to assign them to a specific taxon. The alignment of the 16S rDNA amplicons with the sequences present in the EzBioCloud database (https://www.ezbiocloud.net/, accessed on 16 September 2020) revealed a high level of similarity (>99%) with sequences of the species *Pseudomonas granadensis* and *Rahnella aquatilis*, for formerly *Pseudomonas* sp. A23/T3c and *Enterobacter* sp. BB23/T4d, respectively. ^§^ commercially available (MycAgro; Bretenière, France; http://www.mycagrolab.com/).

**Table 2 microorganisms-09-00426-t002:** Representative Strains of Selected PGPMs Including Corresponding GenBank and RefSeq Accession Numbers for Their Genomes.

PGP Microbial Species	Representative Strain	GenBank Accession No.	RefSeq Accession No.	Reference
*Agrobacterium radiobacter*	K84	chromosome 1/2: CP000628.1/CP000629.1	chromosome 1/2: NC_011985.1/NC_011983.1	[66]
*Azospirillum brasilense*	Sp7	CP012914.1	NZ_CP012914.1	
*Azospirillum lipoferum*	4B	FXBR00000000.1	NZ_FXBR00000000.1	[67]
*Azotobacter chroococcum*	NCIMB 8003	CP010415.1	NZ_CP010415.1	[68]
*Azotobacter chroococcum*	DSM 2286	SRX5354579		
*Azotobacter vinelandii*	CA	CP005094.1	NC_021149.1	[69]
*Bacillus amyloliquefaciens*	DSM 7	FN597644.1	NC_014551.1	[70]
*Bacillus amyloliquefaciens* subsp. *plantarum*; now *Bacillus velezensis*	FZB42	CP000560.1		[71]
*Bacillus atrophaeus* subsp. *globigii*	SRCM101359	CP021500.1	NZ_CP021500.1	
*Bacillus licheniformis*	DSM 13, ATCC 14580	CP000002.3	NC_006270.3	[72]
*Bacillus megaterium*	MSP20.1	CP009920.1	NZ_CP009920.1	[73]
*Bacillus pumilus*	SH-B9	CP011007.1	NZ_CP011007.1	
*Bacillus subtilis* subsp. subtilis	168	AL009126.3	NC_000964.3	[74]
*Bacillus simplex*	SH-B26	CP011008.1	NZ_CP011008.1	
*Burkholderia ambifaria*	MC40-6	chromosome 1, 2, 3: CP001025.1, CP001026.1, CP001027.1	chromosome 1, 2, 3: NC_010551.1, NC_010552.1, NC_010557.1	
*Komagataella pastoris* (*Pichia pastoris*)	ATCC 28485	chromosome 1, 2, 3, 4: CP014584.1, CP014585.1, CP014586.1, CP014587.1		
*Paraburkholderia tropica*	IAC135	chromosome A, B, C, D, E: CP049134.1, CP049135.1, CP049136.1, CP049137.1, CP049138.1	chromosome A, B, C, D, E: NZ_CP049134.1, NZ_CP049135.1, NZ_CP049136.1, NZ_CP049137.1, NZ_CP049138.1	[75]
*Pseudomonas fluorescens*	F113	CP003150.1	NC_016830.1	[76]
*Pseudomonas granadensis*	LMG 27940	chromosome I: LT629778.1	NZ_LT629778.1	
*Pseudomonas putida*	KT2440	AE015451.2	NC_002947.4	[77]
*Rahnella aquatilis*	HX2	chromosome, plasmids PRA1 and PRA2 & PRA22: CP003403.1, CP003404.1, CP003405.1, CP003406.1	NC_017047.1, NC_017060.1, NC_017807.1, NC_017773.1	[78]
*Raoultella terrigena*	NCTC13098	LR131271.1	NZ_LR131271.1	
*Trichoderma harzianum*	CBS 226.95	GCA_003025095.1	GCF_003025095.1	

**Table 3 microorganisms-09-00426-t003:** Pairwise Compatibility among Bacterial Strains and between Each Bacterium and the Yeast *K. pastoris* PP59 Using the Modified Agar Diffusion Method in Nutrient Agar (NA) Plates.

Strain	*A. radiobacter* AR39	*A. brasilense* ATCC 29710	*A. brasilense* NCCB 78036	*A. chroococcum* DSM 2286	*A. chroococcum* LS132	*A. chroococcum* LS136	*A. vinelandii* DSM 2289	*Bacillus* sp. BV84	*B. amyloliquefaciens* BA41	*B. amyloliquefaciens* LMG 9814	*B. licheniformis* PS141	*B. pumilus* LMG 24415	*B. subtilis* LMG 23370	*B. subtilis* LMG 24418	*B. ambifaria LMG 11351*	*B. ambifaria* MCI 7	*B. ambifaria* LMG 11351	*K. pastoris* PP59	*P. tropica* MDIIIAzo225	*Pseudomonas* sp. PN53	*P. fluorescens* DR54	*P. granadensis* A23/T3c	*R. aquatilis* BB23/T4d	*R. terrigena* FS152
*A. radiobacter* AR39																								
*A. brasilense* ATCC 29710	+																							
*A. brasilense* NCCB 78036	+	+																						
*A. chroococcum* DSM 2286	-	-	-																					
*A. chroococcum* LS132	+	+	+	-																				
*A. chroococcum* LS136	+	+	+	-	+																			
*A. vinelandii* DSM 2289	+	+	+	-	+	+																		
*Bacillus* sp. BV84	+	-	+	+	+	+	+																	
*B. amyloliquefaciens* BA41	+	-	+	+	+	+	+	+																
*B. amyloliquefaciens* LMG 9814	+	nc	+	-	+	+	+	+	+															
*B. licheniformis* PS141	+	+	+	-	+	+	+	-	-	-														
*B. pumilus* LMG 24415	+	+	+	+	+	+	+	-	-	-	-													
*B. subtilis* LMG 23370	+	+	+	+	+	+	+	-	-	-	+	-												
*B. subtilis* LMG 24418	+	+	+	+	+	+	+	-	-	-	+	-	-											
*B. ambifaria* LMG 11351	+	+	+	-	+	+	+	+	+	+	+	+	+	-										
*B. ambifaria* MCI 7	+	-	+	-	+	+	+	+	+	+	+	+	+	+	+									
*K. pastoris* PP59	+	+	+	-	+	+	+	+	+	+	+	+	+	+	+	nc	+							
*P. tropica* MDIIIAzo225	+	+	+	-	+	+	+	+	+	nc	+	+	nc	+	+	+	+	+						
*Pseudomonas* sp. PN53	+	+	+	+	+	+	-	-	+	+	+	+	+	+	+	+	+	+	+					
*P. fluorescens* DR54	+	+	+	-	+	+	+	+	+	+	+	+	+	+	+	+	+	+	+	nc				
*P. granadensis* A23/T3c	+	+	+	-	+	+	+	+	+	+	+	+	+	-	+	+	+	+	+	+	+			
*R. aquatilis* BB23/T4d	+	+	+	-	+	+	+	+	+	+	+	+	+	+	+	+	+	+	+	+	+	+		
*R. terrigena* FS152	+	+	+	-	+	+	+	+	+	+	+	+	+	+	+	+	+	+	+	+	+	+	+	

+: compatible; -: incompatible; nc: not clear.

**Table 4 microorganisms-09-00426-t004:** Dual Compatibility Test among Bacteria and Fungi in Potato Dextrose Agar (PDA) Plates.

Bacteria	*T. harzianum* ATCC 48131	*T. harzianum* TH01
*Azotobacter brasilense* ATCC 29710	nc	+
*Azospirillum brasilense* NCCB 78036	-	+
*Azotobacter chroococcum* DSM 2286	-	nc
*Azotobacter chroococcum* LS132	+	+
*Agrobacterium radiobacter* AR39	-	-
*Azotobacter chroococcum* LS163	-	+
*Azotobacter vinelandii* DSM 2289	+	+
*Bacillus* sp. BV84	-	+
*Bacillus amyloliquefaciens* BA41	-	-
*Bacillus amyloliquefaciens* LMG 9814	-	-
*Bacillus licheniformis* PS141	+	+
*Bacillus pumilus* LMG 24415	-	-
*Bacillus subtilis* LMG 23370	-	-
*Bacillus subtilis* LMG 24418	-	-
*Burkholderia ambifaria* LMG 11351	-	-
*Burkholderia ambifaria* MCI 7	-	-
*Komagataella pastoris* PP59	+	+
*Paraburkholderia tropica* MDIIIAzo225	+	nc
*Pseudomonas* sp. PN53	-	+
*Pseudomonas granadensis* A23/T3c	+	+
*Pseudomonas fluorescens* DR54	+	nc
*Ranhella aquatilis* BB23/T4d	-	+
*Raoultella terrigena* FS152	-	+

+: compatible; -: incompatible; nc: not clear.

**Table 5 microorganisms-09-00426-t005:** Qualitative Effect of Bioactive Compounds on Microbial Growth.

Strain	BS1	BS2	BS3	BS4
*Azotobacter chroococcum* LS132	+	+	-	nc
*Azotobacter vinelandii* DSM 2289	+	+	-	nc
*Bacillus* sp. BV84	+	+	-	nc
*Bacillus amyloliquefaciens* LMG 9814	+	+	-	nc
*Bacillus licheniformis* PS141	-	+	-	nc
*Burkholderia ambifaria* MCI 7	+	+	-	nc
*Paraburkholderia tropica* MDIIIAzo225	-	+	-	nc
*Pichia pastoris* PP59	-	+	-	nc
*Pseudomonas* sp. A23/T3c	+	+	-	nc
*Pseudomonas fluorescens* DR54	+	+	-	nc
*Rahnella aquatilis* BB23/T4d	-	+	-	nc
*Trichoderma harzianum* TH01	+	+	+	nc

The symbol “+” means prebiotic effects, i.e., a qualitative increase of growth compared to the control condition; the symbol “-“ indicates no prebiotic effects, i.e., no effect on growth improvement; “nc”: unclear, i.e., samples for which it was not possible to discriminate the positive or negative effect.

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
