# Peer review of "Identification of Beneficial Microbial Consortia and Bioactive Compounds with Potential as Plant Biostimulants for a Sustainable Agriculture"

_microorganisms, 2021, doi:10.3390/microorganisms9020426_

Round 1

Reviewer 1 Report

It is a MS describing the first steps in selection of compatible microorganisms to form consortia and biostimulants, which will promote the growth of four plants.

  • While the experimental part of the study could be accepted as a routine work, there is another part that could be greatly reduced – it contains mainly general statements, project description, and considerations/ suggestions for further studies. For example, the first task of the MS (“Literature survey…”) is without any importance for the reader. It could be reduced and included to another part of the Section Materials and Methods. Similarly, the analysis regarding carriers should be completely reduced: In this MS, experiments with different carriers are missing and therefore this part looks artificial.
  • Some parts of the literature survey are confusing, difficult to follow, and should be restructured. For example, the part on search/selection of microorganisms is based on four plants and field application but the microorganisms’ characteristics many times include information on greenhouse experiments or applied to plants different from the plants-object of this study. In general, some important articles and review papers on microbial inoculants and strategies based on prebiotic, probiotic, postbiotic, synbiotic effectiveness, production and formulation) are not included.
  • Some of the experiments are not well explained. For example, some consortia include bacteria and fungal (Trichoderma) microorganisms. However, biostimulant tests are carried out with the bacterial group (simultaneously-one experimental unit) but with single fungal culture (without its potential compatible bacterial component).
  • The authors mention further combinations with biocontrol agents and arbuscular mycorrhizal fungi but their compatibility test with the microorganisms described in this study is not presented.
  • The characteristics of the four biostimulants should be included in Materials and Methods.
  • The authors should separate the (partial) description of their Project, aims, strategies to follow, analysis of other projects, etc. from the real experimental work. They could describe their project in another (review) paper.
  • The references are not correct. There are references just by the name and article title or without the authors’ names.

Reviewer 2 Report

The authors examined the  PGPMs associated with maize, wheat, potato 
and tomato, and on commercial formulations with potential as agricultural probiotics. The whole work is very good. The novelty high and the presentation clear. I believe that it is a very good work with high scientific level, that offers new and important knowledge in the field. The article is very well written.

Minor comments:

Please explain what do you mean with agricultural probiotics?

Could these agriculture probiotics applied in food fermentation systems. Please clarify and add a sentence in the text.

Would it better to add something about the synbiotics?

Reviewer 3 Report

Tabacchioni et al. studied PGP microbes of wheat, tomato, potato, and maize. They report results from a literature review, metagenome fragment recruitment analysis of PGPMs in public databases, and their own microbial growth experiments. The strength of this paper is the integration of these three different data sources. The limitation is no actual inoculation test and plant growth test, but that will be coming next and there is enough data already in this manuscript to stand alone.  Below I’ve outlined some suggestions for clarifying or improving certain sections, improving the statistics and data visualization, as well as some English edits.

Abstract:

Line 31: may be impaired

Line 39: to help identify

Introduction:

Line 58: arbuscular mycorrhizal fungi

Line 84: synthetic communities

Line 88: frequently increases (no comma)

Line 89: change “improve” to “and improves”

Line 107: delete “object”

Line 108: change “topic” to “the”

The introduction is pretty short, only three paragraphs. This means you could add some more information. In particular, I would like to see more information about the idea that it is hard for the inoculants to establish. I would like to see more information on things that can be done to “prevent rapid decline in the soil” mentioned on line 81 and what exactly about the “chosen formulation” can improve chance of success? Do synthetic communities have more success? How much success? This is key, because if the community you are making is just going to decline after inoculation, then what is the point? This could potentially be a whole separate paragraph.

After reading the whole paper, I will make an additional comment, for the final paragraph of the intro. The introduction should be used also to help organize and give a more complete preview of the paper. In the results there are several main sections, but each is not adequately introduced in the introduction. In line 110 an entire sentence could be devoted to saying how you did a comprehensive genome analysis and that the goal of that was to assess the distribution of the organisms. Instead of simply saying “PGPMs showing the ability to coexist” you could add a bit more detail that it was you in this paper who ran the experiments to determine that and present those results here (that is a major part of the results!). The last parts are good, they mention the goal of functional diversity in the consortia (I suggest using the term “functional diversity”), and then the bioreactive compound tests.

Methods:

Line 121: consisted of

Line 134: was verified

Line 137: change “among which” to “including”

Line 142: Excel

Line 143 to 150: We don’t really need to know your worksheet organization. Delete “Each worksheet…level of abstraction” and start next sentence with “We ranked the scientific articles based on six indicators related to…

Line 153: What was the set of values and was it the same for each indicator? If we don’t know what the values were we can’t assess what a rank >10 actually means.

Line 167: published in

Line 167: in books

Line 171: Can you add a sentence here about what the goals of the fragment recruitment approach are? You may also want to add some more about this in the introduction. In the introduction you do not mention any genomic work, only a very brief and vague mention of acquisition of data.

Line 215: in a given

Line 231: showing overlap

Line 238: Please provide some context here. Why does Trichoderma need a different test that what you already described? Why Trichoderma harzianum? Give some background on Trichoderma harzianum.

Line 253: Please describe what these four compounds are.

Line 258-259: reword, hard to understand.

Results:

Line 278: change “potato studies resulting less represented” to “with potato studies being the least represented.”

Line 278: Instead of just saying potato was least represented, can you just state the number of studies for each of the four crops here?

Line 284: vii should be viii)

Line 288: change “referred to” to “with respect to”

Line 292: were found

Line 294: delete “also”

Line 297: have been developed for potato

Line 298: change “resulted” to “were”

Table 1: center the 61 in the WHEAT row

Line 301: change “brackets” to “parentheses”. Brackets are {}

Line 327-328: points like this need to be made in the introduction to explain why you did the metagenome fragment work in the first place.

Line 338: why are you only mentioning this strain?

Line 357: Arabidopsis is a genus and should be capitalized and italicized. Actually, throughout the paper you should italicize genus and species names appropriately.

Line 383: add a line in the caption to alert readers to the change in the y-axis ordering between the panels.

Line 410: surface area

Table 5. It may be worth writing the species names in the row names and column names to make it easier to follow. You could also only show the lower left of the table as the upper right is just a repeat and not necessary! This would save you space for writing the species names. It would also be helpful to add some color to this to make it more like a heat map to highlight some trends of where there are more - or +. E.g. - red, + blue, nc white. This would make it easier to see which species have more - or + or nc.

Line 478: again, there was no explanation of why you were specifically testing for compatibility with T harzianum. This needs to be explained clearly early in the paper (see comment in methods).

Line 538: bacterium

Line 590: you need to report the ANOVA results in order to say significant increase.

Figure 5: This is not the best way to graph this in my opinion. Instead of bars you could use satellites (points with error bars) connected by lines so there would be 6 lines. The x-axis would only have 3 locations on it (24, 48, 72) and the satellites would be spread vertically across the y-axis at each time point (they could be slightly dodged for legibility if necessary). I hope this makes sense (it would look something like the answer here https://stackoverflow.com/questions/59195398/plot-time-series-in-r-ggplot-using-multiple-groups but you would have only 3 points and also show error bars). Also, now that I am seeing these data, I think some more information is needed on how you set up the ANOVA model. It should be a repeated measures ANOVA. We see that they all increase over time, so I think what we are interested in is a significant interaction, meaning that the magnitude of increase for some is greater.

Conclusions:

Line 633: and nature

Check the font, some words appear to be written in different font.

Line 654: change “permit” to “enable us”

Line 656: delete “to” after hence

Round 2

Reviewer 1 Report

Now the MS is better and describes the real experimental work.